# Saliva Molecular Testing for SARS-CoV-2 Surveillance in Two Italian Primary Schools

**DOI:** 10.3390/children8070544

**Published:** 2021-06-24

**Authors:** Daniela Carmagnola, Gaia Pellegrini, Elena Canciani, Dolaji Henin, Mariachiara Perrotta, Federica Forlanini, Lucia Barcellini, Claudia Dellavia

**Affiliations:** 1Department of Biomedical, Surgical and Dental Sciences, Università degli Studi di Milano, Via Mangiagalli 31, 20133 Milano, Italy; daniela.carmagnola@gmail.com (D.C.); gaiapellegrini.perio@gmail.com (G.P.); elena.canciani@unimi.it (E.C.); mariachiara.perrotta1997@gmail.com (M.P.); claudia.dellavia@unimi.it (C.D.); 2Department of Pediatrics, Ospedale dei Bambini V. Buzzi, University of Milan, 20133 Milano, Italy; federica.forlanini@unimi.it (F.F.); lucia.barcellini@unimi.it (L.B.)

**Keywords:** molecular salivary testing, school surveillance, COVID-19

## Abstract

Background: No evidence has so far proven a significant role of schools in SARS-CoV-2 transmission, while the negative effects of their closure on children and adolescents are well documented. Surveillance, by means of frequent students and staff testing, has been advocated in order to implement school safety. Our aim was to report the results of a school surveillance program for the early detection of SARS-CoV-2 infection in pre- and asymptomatic subjects, by means of molecular salivary testing (MST). Methods: School surveillance in two schools in Milan, Italy, was carried out for six weeks. Each participant received a saliva collection kit, to be self-performed. Results: 401 students and 12 teachers were enrolled, and 5 positive children in 5 different classes were observed. All the cases were asymptomatic. Their nasopharyngeal swab was positive on the same day in four cases, while in one case it resulted negative on the same day and positive 3 days later. In one positive case, the whole family was set under surveillance. The positive child did not develop symptoms and no family member was infected. Conclusions: MST might represent an efficient way to actively survey communities in order to detect asymptomatic cases, thus limiting SARS-CoV-2 transmission.

## 1. Introduction

Taking reasonable decisions concerning when and how to open and close schools during the COVID-19 pandemic, has been an issue worldwide. On the other hand, the negative effects of long-term school closure and online education on children and adolescents are well documented, especially concerning those in fragile and disadvantaged conditions [1,2,3,4]. 

In general, most countries have chosen to limit school access and moved learning online, in the name of a precautionary approach, despite continuous efforts to improve safety in the scholastic setting [5]. Modelling studies have, for example, predicted a 2 to 4% death reduction, thanks to school closures, but so far the actual benefits of such measure on death control are not documented [6]. Furthermore, it is not yet clear how children may actually affect or be affected by the SARS-CoV-2 infection [7]. According to data from the screening study in the city of Vo’ Euganeo (Padua, Italy), where 70% of the population was tested after the first Italian wave in 2020, 2.6% resulted positive and among those was no child below 10 years of age, even when living in households with positive members [8]. Also concerning the amplification of SARS-CoV-2 transmission, no evidence has so far proven a significant role of school attendance [9].

As a matter of fact, short periods of school presence have been allowed after implementing physical distancing, ventilation, proper respiratory etiquette, the use of masks and regular handwashing. Further, strict screening, diagnostic and tracing protocols for both students and staff have been introduced, thus contributing to community testing, tracing and tracking [1,10]. 

Nevertheless, at every COVID-19 wave, stay-at-home measures, including school attendance limitations, have been introduced. The burden of school closures has weighed mainly on the students’ families, who have, in the absence of clear evidence in favor of closing, often claimed they be re-opened.

In Italy, in late October 2020, as COVID-19 cases in the general population increased, in order to reduce school-related testing burden and influenced by the hypothesis that schools might act as clusters or contribute to amplify the overall infection rates, distance learning for older pupils (age > 16) was adopted. Such measure was extended, a few days later, to middle schools (age > 12), until 30 November. Until the end of January 2021, in general only children 0 to 13 years of age were allowed in presence at school. 

Such context of isolation has raised reasonable worries about the potential negative implications on children’s education, psychological and social life. 

Since February 2021, concomitantly with a decrease in the effective reproduction number R, all schools have eventually opened, without a proper safety protocol. 

The only measure proposed for SARS-CoV-2 transmission control was the immediate exclusion of the child from the class on the onset of COVID-19 symptoms, and his/her referral to a pediatrician for evaluation and eventually, if needed, for a molecular nasopharyngeal swab (MNPS). While waiting for the MNPS result, the student is placed under quarantine. In the case of positivity, the whole class is placed under quarantine for 10 to 14 days. Both the positive student and classmates can be readmitted to school after at least 10 symptomless days followed by a negative MNPS, or 14 symptomless days without MNPS.

Surveillance by means of frequent students and staff testing has been often advocated in order to further implement school safety, and therefore attendance. MNPSs are considered the gold standard tool for identifying SARS-CoV-2 infection [11], but they are invasive, and require health personnel and logistic resources to be performed on a large scale, making them inadequate for frequent testing or surveillance. Other diagnostic methods include nasopharyngeal antigenic swabs (NPaS) and salivary tests, both antigenic and molecular. Molecular salivary testing (MST) has proven to be a reliable tool for SARS-CoV-2 detection [12,13]. In particular we evaluated the MST performance in 192 adults, finding a concordance of 0.85 and a k coefficient of 0.69 between MNPS and MST, indicating a substantial concordance (Cohen′s unweighted Kappa, k). In asymptomatic subjects, concordance rose to 0.96 with k equal to 0.83, indicating an almost perfect concordance. In a pediatric population, including 109 children, the overall concordance was 0.94 and k was 0.81 [14]. Comforted by the results, in particular on asymptomatic subjects, we decided to evaluate MST in a surveillance program in two primary schools. 

The aims of the current observational study were as follows: (i) to report data from a school surveillance program for the early detection of SARS-CoV-2 infection in 6 to 11 years old children attending primary school, by means of MST; (ii) to describe the progression of infection in a positive early detected child and the dynamics of transmission in his family, and (iii) to discuss the contribution of active surveillance at school to further improve its safety and avoid closures. 

## 2. Materials and Methods

A school surveillance program was started in November 2020 in two primary schools (corresponding approximately to grades 1 to 5 in other systems) in Milan, Italy, and was carried out for 6 weeks. The project’s design was announced by email to the school personnel and students’ families by the school principal two weeks ahead of its start. The target of the study was the school children, but the program was extended to the main class teachers in order to further understand the dynamics of SARS-CoV-2 transmission within the class. 

Participation was voluntary. The study was approved by the local ethical committee (Comitato Etico Interaziendale Milano Area A, N. 0050308). All participants/legal representatives signed an informed consent, in accordance with the Declaration of Helsinki, and received detailed information on how to self-collect saliva. The principals of the 2 schools were interviewed concerning previous COVID-19 cases in their schools.

Briefly, each participant received a tube containing a dental roll for saliva collection once a week, to be self-performed in the morning. After thorough hand cleaning and at least 30 min after eating, drinking or tooth brushing, each participant placed a dental roll in the mouth, namely, in the lower vestibular space next to the premolar–molar area and under the tongue in the Wharton duct area (Figure 1), for 3–4 min. When duly soaked, the roll was placed back in the tube and taken to school, where all samples were collected and delivered to a university lab for analysis within 24 h. 

The first sampling was guided by a team of researchers at school. Subsequently a dedicated movie, illustrating the passages necessary to correctly collect saliva samples was sent to all children families in order to self-perform the test at home. The movie was also uploaded onto the school’s website. 

The MST was adapted from the SalivaDirectTM protocol developed by the Yale School of Public Health [15]. Saliva was recovered from the tube under sterile conditions using a 10 mL syringe to squeeze the cotton roll. Collected saliva was processed as recommended by Vogels et al., extracting nucleic acid with proteinase K and heat using a modification of the thermal cycler profile, as follows: 5′ at 95 °C followed by 5′ at 4 °C [14,15]. Five microliters of the processed saliva were used directly for qPCR both for N1 (FAM, BHQ-1 labeled probe) and for RP (Cy5, BHQ-1) primers/probes published by the Center for Disease Control and Applied Biosystem 7500 Fast instrument [16]. Samples were considered positive upon detection of N1 (Ct < 40). Invalid samples were assessed by RP (Ct > 35).

In the present paper we report the saliva molecular testing results from the 6 weeks surveillance program and data. In case of positive subjects, all their households were offered the possibility to be included in a surveillance program. 

## 3. Results

In this longitudinal study, we surveilled weekly children and teachers of two schools for a six week period. Twelve teachers (out of 60, 20%) took part in the study, and not one of them resulted positive. Out of 578 children attending the 2 schools, 401 eventually took part in the study (69%). The mean age of the students was 8 years (range: 6 to 11) and 51% were male.

Table 1 shows the analyzed samples at the different timepoints, since the school attendance was irregular, mainly because of quarantine. 

Concerning COVID-19 history in the two schools, the school principals reported that six classes were under quarantine, as four students and four teachers had previously tested positive, and this contributed to the initial low number of participants. 

As the study proceeded, the classes were released from quarantine, the involvement and compliance of the families grew, and the number of participants grew. The last week coincided with the last school day before the Christmas holidays, and, as stay-at-home measures had been announced by the Italian government starting from the 24 December, many families anticipated their departure towards their vacation locations. On average, about 10% of the saliva samples could not be processed each week due to the following two reasons: either the family forgot to bring the sample to school, or the sample was too dry to provide an adequate amount of saliva. In the surveillance period, a total of 1619 MST were analyzed.

During the six week program, five positive children in five different classes were observed (one at week one, two at week two, two at week three), and no teachers. All the cases were totally asymptomatic at the time of diagnosis. The MNPS was positive on the same day in four cases, while in one case it resulted negative on the same day and positive 3 days later. The classmates were placed under quarantine for 14 days, and were not farther tested by any means. No classmate or teacher developed any symptoms. Figure 2 shows the temporal evolution of the infection. 

The whole families of the positive children were offered to be monitored by MST every three days, but only two families agreed, and one only complied in providing all the required data. 

In one family, the MST and MNPS of the positive child mismatched on day one, but matched 3 days later. In this family, the sister of the positive student resulted positive to MST and MNPS. The family did not comply further in providing salivary samples. One other family agreed to be monitored by MST and complied with providing all the required data. This household included a 49-year-old mother, a 52-year-old father, a 12-year-old son and the 6-year-old student himself. None of them reported pathologies or were on medications. Their 6-year-old child had been tested the week before (T1, Table 2) and resulted negative. 

The family underwent MST and MNPS on the same day (T2, Table 2) of the positive MST and MNPS of their 6-year-old, and no one else resulted positive (the results were provided within 5 h). The family started isolation and were recommended to wear FFP2 masks, to frequently ventilate all rooms, and to isolate the child as extensively as possible.

Saliva was analyzed about three times a week for the following two weeks, and always proved positive for the child and negative for the rest of the family. The cycle threshold (Ct) values are reported in Table 2, among brackets. The mother, father, and their 12-year-old son underwent a serological capillary ELISA test at day 5 and at day 20 after T2, and they all had negative results [17]. The 6-year-old son did not comply with the serological test. No one developed any specific signs or symptoms.

According to Italian regulations, the 6-year-old child underwent a final MNPS 10 symptomless days after the first MNPS. As this resulted negative, he could be readmitted at school after 14 days. His schoolmates had been isolated as “close contacts” from the day of his positive MNPS, for 14 days. As no one developed any symptoms, they could all be readmitted at school with no further measure. The family ended the quarantine after 14 symptomless days. The child’s classmates involved in the study all had negative saliva samples at both time 0 and 1, as well as in the following weeks.

## 4. Discussion

The results of the present study show that MST can be a well-tolerated, precise, simple, cheap and useful tool for the diagnosis of SARS-CoV-2 infection in children, and for active school surveillance. MST has proven to be able to detect asymptomatic cases early.

Our results are in agreement with those by Wyllie et al. (2020) [13], who reported that MST seems to be indicated for the early detection of pre- and asymptomatic subjects, even before the MNPS is able to identify a positivity. Furthermore, MST is less invasive than a nasopharyngeal swab, and therefore it can be repeated frequently and self-performed at home or in any location, and reduce the exposure risk of health professionals and the need for dedicated spaces; it requires about the same analysis equipment and time as MNPS and it costs as much, minus the expenses related to sampling personnel and logistics [15]. Its results can be provided quite quickly, within 24 h. The abovementioned characteristics make MST the optimal tool for the early detection of SARS-CoV-2 infection in children, and for active school surveillance programs including students and staff.

Despite MST, as well as antigenic and molecular nasopharyngeal swabs, being a cheap tool, active school surveillance on a large number of students and staff implies important economical costs. On the other hand, closing schools has resulted in other types of costs. Investing in surveillance programs might contribute to implement a school’s safety, necessary to keep schools open, and hence reduce the well-documented disadvantages and burden caused by their closures on students and families [18]. Concerning the consequences of schools closures on the well-being of children and adolescents, the literature seems to agree on the fact that underprivileged children from disadvantaged families, and disabled children, are the ones who suffer the most, in many respects, from not attending school, while students with culturally and socio-economically advantaged backgrounds might even develop and expand skills and competences, in terms of communication possibilities and device exploitation. School closure might therefore contribute to increase the already existing gap between advantaged and disadvantaged children and adolescents, and recent studies confirm that this has been the case during the COVID-19 pandemic [19,20,21,22,23,24].

A review including 22 studies has concluded that school closure and lockdown, due to the pandemic, may cause adverse effects on child health and well-being in the short and probably long term [25]. A literature review has shown how isolation has actually negatively affected the mental health of children and teens [26]. Similar findings have been reported by Jimenez et al. (2020), who demonstrated that quarantine may represent an important source of stress, additionally to other psychological manifestations, such as bad mood, depression, anxiety and even delirium, as well as in studies, for example, from China and Japan [27,28].

In the present study, conducted during a COVID-19 wave in Italy, we detected only 5 positive students out of 401 children, and no positive teachers, during a six week surveillance period. All were asymptomatic.

Many studies report that COVID-19 in children tends to be symptomless or minimally symptomatic, and some researchers suggest a low circulation of SARS-CoV-2 infection in pediatric communities, including schools [29,30].

Some studies have advocated school closures or attendance limitations as a measure to decrease SARS-CoV-2 transmission, due to the large number of COVID-19 cases among school-aged children and adolescents in the periods in which schools were open [31]. It must be noticed though that even in the absence of regular surveillance programs, the scholastic setting is generally regulated by strict protocols that aim at detecting early signs and symptoms that are suggestive of COVID-19, often leading to testing and contact tracing. In this respect, the scholastic communities are more frequently tested than the rest of the population. In a study analyzing data from different institutional Italian databases, the authors observed that SARS-CoV-2 incidence among students was lower than in the general population, while among teachers while among teachers it was comparable to the one observed in the general population of similar age. Further, school clusters could be identified in only 5% to 7% of the institutes [32].

The school surveillance weekly protocol we carried out allowed us to detect five positive asymptomatic cases of SARS-CoV-2 infection. Therefore, the children and their families were isolated, and contact tracing measures were started immediately. In the absence of surveillance, the children would have attended school as usual, and their families would have not adopted strict distancing and protective home measures had they not become aware of their children’s infection. It is known that children are less susceptible to, and rarely present, severe forms of COVID-19 [8,33]. 

Household transmission to more fragile subjects remains an open issue. A recent American study has investigated SARS-CoV-2 transmission in 58 households of individuals with coronavirus disease. According to the authors, “children and adults had similar secondary infection rates, but children generally had less frequent and severe symptoms. In two states early in the pandemic, we observed possible transmission from children in approximately one-fifth of households with potential to observe such transmission patterns” [34]. 

Frequent MST has allowed the monitoring of the Ct values of one positive child. Such values defined a line that reached its peak at day nine (2 days after the infection had been detected), and had since then moved towards higher values. As we set cut-off values for positivity at 40 cycles, the child was considered negative at day 20 (13 days after diagnosis), and the MNPS performed on the same day was concordant. 

Since the MST can be repeated frequently, such as three times a week, it might be suggestive of the proper time for the positive child to safely get back into the community. 

Furthermore, close contacts in general need to be isolated for 14 asymptomatic days, which generally means two weeks absence from school or work in cases where one school- or workmate is positive. Considering that the incubation time of COVID-19 is about 3/9 days, a procedure in which close contacts who present three consecutive negative saliva results within a week might be considered safe to go back to school or work, might be of help in reducing the isolation period of non-infected individuals, including its negative social, psychological and economic consequences [35].

This study has set the basis for considering MST as a useful and simple tool for school surveillance, with the main advantages of (1) early detection of pre/asymptomatic cases, (2) frequent testing that allows the reduction in the isolation period, and (3) understanding the dynamics of virus transmission within the school class. 

However, some limitations, such as the small number of students and schools, the short surveillance time, and the moderate compliance (10% of not-detectable samples), have to be mentioned. Further, in the current study, we found a moderate–high value of adherence rate in children and moderate–low in teachers. These data could be explained by the experimental nature of the program, requiring the positive MST confirmation by means of MNPS. All these limitations may be overcome, thanks to the recent regulation of MST by the Italian government. On 15 May 2021, the Italian Ministry of Health validated MST as an option for the detection of SARS-CoV-2 infection in particular cases/setting, as for example in communities including children. As a consequence, some local sanitary authorities are assessing the feasibility of a school surveillance system based on self-performed MST on a larger scale than ours. Logistical aspects, such as samples picking up and delivering to certificated laboratories for analysis, are in fact crucial to guarantee an effective model of COVID-19 school surveillance. Once this phase will be completed, surveillance of children below 12 years of age, together with vaccines for >12 years old students, might help in improving school safety and reduce the need for closures in case of future waves. 

## 5. Conclusions

In conclusion, all possible efforts and investments should be made to allow students from all grades to attend school in presence. Increasing school safety and stressing the importance of safe behavioral measures, rather than closing schools, seems the most rational approach to cope with COVID-19, while keeping schools open and avoiding the abovementioned dramatic consequences of school closures on children and their families [2,36,37,38]. 

So far, schools, provided with strict COVID-19 control protocols, have proven to be quite a safe place. MST for active surveillance programs might represent an efficient way to further contribute to preserve or minimize school closures. The investment and effort in repeated sampling and analysis could be counterbalanced by the possibility of early detection of a- or pre-symptomatic students and their contacts testing, thus narrowing quarantines, as well as their consequences, to what is strictly necessary. Not every sanitary system would be able to test the whole scholastic population once a week, but epidemiological analysis might come handy in identifying the riskiest age groups, also by defining whether the positivity observed in some age groups springs from the scholastic setting or the opposite, i.e., from extra-scholastic gatherings in the absence of any surveillance.

## Figures and Tables

**Figure 1 children-08-00544-f001:**
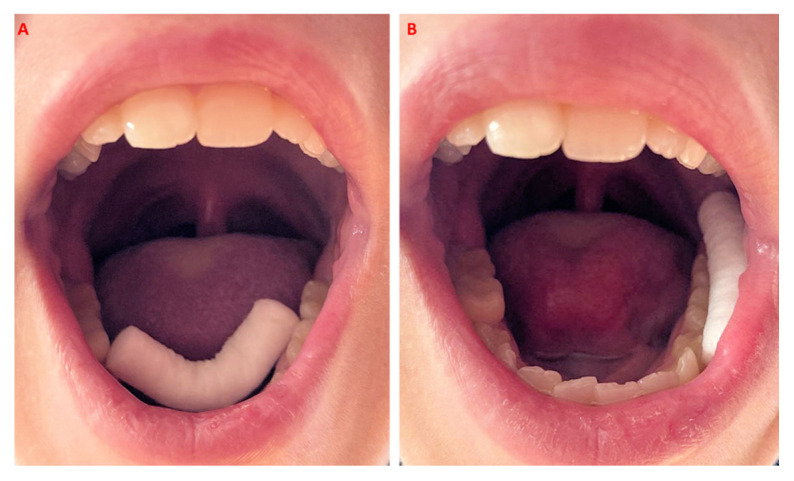
Picture of the cotton roll inserted under the tongue to collect and promote saliva production (**A**) and in the lower vestibular space (**B**).

**Figure 2 children-08-00544-f002:**
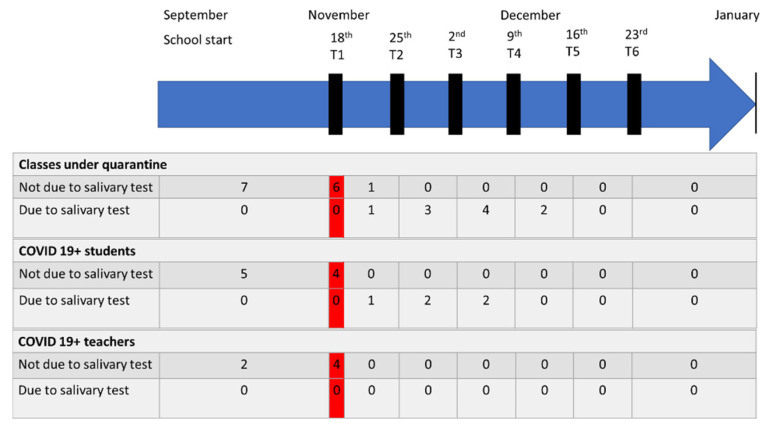
Overview of the COVID-19-related quarantine trend during the first school term. The graphic shows the number of classes, students and teachers starting and ending the 14 days quarantine. In the red boxes, a photo of the situation at the beginning of the surveillance. At the left of the red boxes the quarantine and transmission status before the start of the surveillance, and at the right the status after.

**Table 1 children-08-00544-t001:** The table reports the number of studied subjects at the different timepoints, the percentage of analyzed samples, not detectable samples, and samples positive to MST (molecular salivary testing).

	T1	T2	T3	T4	T5	T6
Total adhesions (2 schools)	333	361	373	389	401	401
Analyzed (%)	203 (61)	271 (75)	290 (77)	297 (76)	310 (77)	248 (64)
Not detectable MST %	8	11	11	9	11	8
Positive MST (%)	1 (0.5)	2 (0.7)	2 (0.7)	0 (0)	0 (0)	0 (0)

**Table 2 children-08-00544-t002:** The table shows the Ct values of the 6-year-old child in the two weeks monitored. MNPS = molecular nasopharyngeal swab; MST = molecular salivary testing; P = positive; N = negative.

	T1	T2	T2 + 2 Days	T2 + 4 Days	T2 + 6 Days	T2 + 8 Days	T2 + 11 Days	T2 + 13 Days	T2 + 15 Days
MNPS		P						N	
MST	N	P (34.3)	P (28.9)	P (36.4)	P (37.4)	P (32.3)	P (38.9)	N (40.5)	N

## Data Availability

The data that support the findings of this study are available from the corresponding author upon reasonable request.

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
