# Peer review of "Saliva Molecular Testing for SARS-CoV-2 Surveillance in Two Italian Primary Schools"

_children, 2021, doi:10.3390/children8070544_

Round 1

Reviewer 1 Report

First of all I wanted to congratulate the authors for the article, since I find it very interesting to provide data on surveillance tests for Sars-CoV-2 in schools as a method to avoid closures and it is very useful for public health. However, the article has some shortcomings that need to be addressed:

  • It would be convenient to extend the Introduction with a deeper justification, for example, talking about the psychological impact of quarantine (depression, anxiety, stress) on young people. You can include, for example, this study ("Psychological impact of COVID-19 confinement and its relationship with meditation"). This study was published in the journal:

https://www.mdpi.com/1660-4601/17/18/6642

  • In the Results section: why did you include the data from the entire sample that you collected at each point in time rather than the participants who are assessed at all times, such as a longitudinal study?

Reviewer 2 Report

Excellent straightforward account of saliva testing of primary school aged children, for SARS-CoV-2 infection. It was very good to see such young children actively participating in a research project. I hope that they will have opportunity to be involved in disseminating the results. 

You have acknowledged the limitations of the study, and while response rate could maybe be improved, it is probably realistic in the 'real world' to have families who cannot comply every week, due to external (or internal) pressures. 

Reviewer 3 Report

Firstly, I commend your effort for conducting such a study at this difficult time and with a vulnerable population.

The introduction is sound. It provided key information about current overview of school closure due to the pandemic, countries response to schools, and some of the measures been implemented to minimise school transmission. The rational for the study is justifiable considering the fact that limited evidence has so far demonstrated a significant role of school attendance in the amplification of covid transmission.

In the introduction, I thought some key statistics (locally or globally) are important about covid in children and/or reported school transmissions if any. That would further strengthen the need for and importance of the study.

The research aims were clear, including to report data from school surveillance program for the early detection of covid infection in primary school children, progression of infection in a positive early detected cases/transmission families and to examine active surveillance at the schools.

The method section provided some information about how the study was conducted. However, method of analysing the data was not presented. Also, it is good to make the participants clear at the method. Initially I thought only children constitute the sample but later come to realise it included some teachers. Such information may allow for possible replication of the study if needed.

Results; could you be clear about the number of teachers participated and their covid status? Line 95 mention none of the teachers who participated resulted in a positive result while line 103 says 4 teachers initially tested positive. I may have missed the connection but would be good to simplify.

While the discussion is detailed enough to cover the key findings, some claims are made without evidence/reference. For instance, line 195-198 ‘In a study analyzing data from different institutional Italian databases, the authors observed that SARS-CoV-2 incidence among students was lower than in the general population while among teachers it was comparable to the population of similar age and school clusters could be identified only in 5% to 7% of the institutes’. There were couple of other strong statements like that, which need to be backed up in the discussion. Please check and insert the references accordingly.

The fact that studies in this area among school children is limited, you may consider mentioning a few strengths of the findings before the limitations.

Good work. I enjoyed reading the manuscript.

Regards
